# CIGB-258 Exerts Potent Anti-Inflammatory Activity against Carboxymethyllysine-Induced Acute Inflammation in Hyperlipidemic Zebrafish via the Protection of Apolipoprotein A-I

**DOI:** 10.3390/ijms24087044

**Published:** 2023-04-11

**Authors:** Kyung-Hyun Cho, Hyo-Seon Nam, Ji-Eun Kim, Hye-Jee Na, Maria del Carmen Dominguez-Horta, Gillian Martinez-Donato

**Affiliations:** 1Raydel Research Institute, Medical Innovation Complex, Daegu 41061, Republic of Korea; 2LipoLab, Yeungnam University, Gyeongsan 38541, Republic of Korea; 3Center for Genetic Engineering and Biotechnology, Ave 31, e/158 y 190, Playa, La Habana 10600, Cuba

**Keywords:** apolipoprotein A-I (apoA-I), carboxymethyllysine (CML), hyperinflammation, CIGB-258 (Jusvinza^®^), zebrafish, high cholesterol diet

## Abstract

Inflammation and atherosclerosis are intimately associated via the production of dysfunctional high-density lipoproteins (HDL) and modification of apolipoprotein (apo) A-I. A putative interaction between CIGB-258 and apoA-I was investigated to provide mechanistic insight into the protection of HDL. The protective activity of CIGB-258 was tested in the CML-mediated glycation of apoA-I. The in vivo anti-inflammatory efficacy was compared in paralyzed hyperlipidemic zebrafish and its embryo in the presence of CML. Treatment of CML induced greater glycation extent of HDL/apoA-I and proteolytic degradation of apoA-I. In the presence of CML, however, co-treatment of CIGB-258 inhibited the glycation of apoA-I and protected the degradation of apoA-I, exerting enhanced ferric ion reduction ability. Microinjection of CML (500 ng) into zebrafish embryos resulted in acute death with the lowest survivability with severe developmental defects with interleukin (IL)-6 production. Conversely, a co-injection of CIGB-258 or Tocilizumab produced the highest survivability with a normal development speed and morphology. In hyperlipidemic zebrafish, intraperitoneal injection of CML (500 μg) caused the complete loss of swimming ability and severe acute death with only 13% survivability 3 h post-injection. A co-injection of the CIGB-258 resulted in a 2.2-fold faster recovery of swimming ability than CML alone, with higher survivability of approximately 57%. These results suggest that CIGB-258 protected hyperlipidemic zebrafish from the acute neurotoxicity of CML. Histological analysis showed that the CIGB-258 group had 37% lower infiltration of neutrophils in hepatic tissue and 70% lower fatty liver changes than those of the CML-alone group. The CIGB-258 group exhibited the smallest IL-6 expression in the liver and the lowest blood triglyceride level. CIGB-258 displayed potent anti-inflammatory activity in hyperlipidemic zebrafish by inhibiting apoA-I glycation, promoting rapid recovery from the paralysis of CML toxicity and suppression of IL-6, and lowering fatty liver changes.

## 1. Introduction

Inflammation is a series of complex pathological events initiated by infection or cell damage with non-specific attack [1]. Hyperinflammation is associated with many pathological processes of critical diseases, such as cancer [2], atherosclerosis [3,4], neurological disorder [5], Coronavirus disease 2019 (COVID-19) [6], and autoimmune diseases [7]. A cytokine storm, which is an abnormally elevated level of inflammatory cytokines, such as tumor necrosis factor (TNF)-α and interleukin (IL)-6, by the innate immune system, is frequently accompanied by acute hyperinflammation in patients with infectious diseases and autoimmune diseases [7]. Because the cytokine storm is closely linked with multi-system organ failure and acute death, many treatment approaches, including corticosteroids, intravenous immunoglobulin, and Tocilizumab, have been applied to alleviate cytokine storm [8]. However, there is no effective and specific drug to treat hyperinflammation and cytokine storm because of diverse clinical syndromes and cytokines depending on inflammatory conditions.

Inflammation and atherosclerosis are intimately associated via the production of dysfunctional high-density lipoproteins (HDLs) and modification of apolipoprotein (apo) A-I. High-density lipoproteins exert anti-inflammatory activity as a humoral part of innate immunity with anti-viral, anti-bacterial, and anti-parasitic activity [9]. HDLs have a potent anti-inflammatory and anti-cancer activity to suppress tumor-promoting inflammation [10]. ApolipoproteinA-I (apoA-I), a major HDL protein with 28 kDa, also displayed independently potent anti-inflammatory activity on adipocytes [11,12]. HDL/apoA-I activated CD8+ T cells, anti-inflammatory macrophages, and neutrophils to exert immunomodulatory activities [13]. HDL/apoA-I also suppressed the maturation, activation, and chemotaxis of dendritic cells. In addition, it inhibited T cell proliferation and increased CD4+ T cell survival, resulting in immunosuppression [14]. The administration of apoA-I in experimental autoimmune uveitis (EAU) mice ameliorated EAU by modulating T_eff_/T_reg_ balance through scavenger receptor B-I [15]. ApoA-I can bind bacterial endotoxin, lipopolysaccharide (LPS), and lipoteichoic acid (LTA) to protect the progression of sepsis via the lower activation of toll-like receptor (TLR)-4 [16,17]. Sepsis is an important cause of cytokine storm, involving increased TNF, IL-1β, and IL-6 production through proinflammatory cells and sepsis pathology [18]. Based on established evidence, enhancing HDL/apoA-I in quantity and quality is an effective strategy to suppress the pathological processes of infection, immunity, and autoimmunity [19]. The anti-inflammatory roles of reconstituted HDL/apoA-I are exerted by immune cell modulation, such as inhibiting adhesion and chemotaxis in monocytes, neutrophils, and eosinophils [14,20].

Our preceding paper [21] reported that CIGB-258, an altered peptide ligand (27 amino acids, MW = 2987) derived from heat shock protein HSP60, protected HDLs from carboxymethyllysine (CML)-induced glycation and ferrous ion-mediated proteolytic degradation via stabilization of the protein structure and antioxidant ability of HDLs. The anti-inflammatory activity of CIGB-258 might have originated partially from the structural and functional stabilization of HDLs against CML-induced glycation of HDLs and modification [21]. Although the CML is a relatively stable advanced glycated end (AGE) product, many previous papers suggested its neurotoxicity and oral toxicity [22,23,24]. The CML can bind to tissue proteins [25] and accumulate in animal tissues after 12 weeks of oral administration via glycoxidation and lipid peroxidation [26]. The CML-modified proteins activated a proinflammatory response and vascular complications via loss of receptor of AGE binding ability [27].

In our preceding paper, as an in vivo test, intraperitoneal injection of CML (250 μg) into normolipidemic zebrafish caused acute paralysis and death with a loss of swimming ability [21]. A co-injection of CIGB-258 displayed anti-inflammatory activity against the acute toxicity of the CML in paralyzed zebrafish with faster recovery of swimming ability and survivability. Furthermore, regarding anti-inflammatory activity, CIGB-258 showed similar features to an IL-6 inhibitor (Tocilizumab) than a TNF-α inhibitor (Infliximab) to protect zebrafish and embryos from acute death by CML treatment. Our previous paper proved that CIGB-258 showed potent anti-inflammatory activity in the normolipidemic state [21]. However, inflammation is exacerbated more by hyperlipidemia and obesity, particularly in COVID-19 patients [28]. Non-survivors displayed elevated serum C-reactive protein, ferritin, and IL-6 levels upon hospital admission [29,30]. The intravenous administration of CIGB-258 (1 or 2 mg for every 12 h) showed the therapeutic potential to treat hyperinflammation of patients with COVID-19 with a remarkable decrease in cytokine levels present in “cytokine storm”, such as IL-6 and TNF-α [31]. These results are consistent with those reported in preclinical studies and a phase I clinical trial with rheumatoid arthritis patients [32,33,34].

Based on our previous report [21], it is necessary to test the anti-inflammatory activity in a hyperlipidemic animal model and to compare it with other protein drugs and monoclonal antibodies. However, there have been no reports comparing the abilities of the anti-inflammatory IL-6 inhibitor (Tocilizumab) and TNF-α inhibitor (Infliximab and Etanercept) to inhibit CML-induced inflammation in a hyperlipidemic state.

As a series of efficacy studies, therefore, the current study was designed to confirm the anti-inflammatory efficacy of CIGB-258 using CML-injected hyperlipidemic zebrafish. This study compared the anti-inflammatory activity of CIGB-258 in hyperlipidemic zebrafish and its embryo and anti-glycation activity to protect lipid-free apoA-I with commercially available monoclonal antibodies, such as Infliximab, Etanercept, and Tocilizumab.

## 2. Results

### 2.1. Glycation and Proteolytic Degradation of HDL by CML Treatment

As shown in Figure 1A, treatment of CML into HDL_3_ caused a remarkable increase in the glycation extent, in a dose-dependent manner from 100 μM to 400 μM (final) during 144 h at 37 °C under 5% CO_2_, up to 29% increase in yellowish fluorescence by CML 400 μM (final) treatment than HDL_3_ alone. Under the same conditions, a fructose treatment (final 400 μM) caused only an 8% increase in the glycation extent, suggesting that CML showed more reactivity to produce a more AGE product. After 144 h incubation, SDS-PAGE analysis showed that the proteolytic degradations of apoA-I occurred by CML treatment in a dose-dependent manner, as shown in Figure 1B. Up to 45% of the band area of apoA-I disappeared after the CML treatment (final 400 μM) at 144 h, while the fructose-treated (final 400 μM) HDL_3_ showed only a 7% loss of the apoA-I band. These results suggest that the CML has remarkably higher reactivity than fructose to cause glycation of HDLs in both fluorescence intensity and proteolytic degradation of apoA-I.

### 2.2. Anti-Glycation Activity of CIGB-258

Treatment of lipid-free apoA-I with CML produced an eight-fold more AGE product than apoA-I alone, as shown in Figure 2A. Co-treatment of 10 μM and 100 μM CIGB-258 caused 17% and 25% inhibition of the glycation in apoA-I, respectively. The glycated apoA-I by CML (lane 2) showed a smaller band size and intensity, approximately 36% less than apoA-I alone (Figure 2B), indicating that the glycation of CML caused the proteolytic degradation of apoA-I. Conversely, co-treatment of CIGB-258 prevented apoA-I from degradation in a dose-dependent manner. The CML treatment caused the loss of up to 36% of the apoA-I band (lane 2) compared to apoA-I alone (lane 1). In contrast, the co-treatment of CIGB-258 protected the loss of the apoA-I band in a dose-dependent manner. The band intensity was increased up to 1.8-fold by the co-presence of 100 μM of CIGB-258 (lane 5), compared with apoA-I + CML (lane 2).

### 2.3. Stabilization of apoA-I by CIGB-258

Treatment of apoA-I with CML caused a 3.1 nm red-shift of the wavelength maximum fluorescence (WMF) during 48 h incubation, as shown in Figure 3A; apoA-I alone showed 343.0 nm, while CML-treated apoA-I showed 346.1 nm of WMF. Conversely, the co-presence of CIGB-258 resulted in a 0.2 nm, 0.7 nm, and 2.9 nm blue shift of WMF at the final 1, 10, and 100 μM treatments, respectively. These results suggest that the CML treatment induced exposure of Trp in apoA-I toward the aqueous phase via a red-shift of WMF. The co-treatment of CIGB-258 caused the movement of Trp to the nonpolar phase via the putative interaction of CIGB-258 and CML to exert anti-glycation activity.

As shown in Figure 3B, a comparison of the ferric ion reduction ability (FRA) showed that the CML-treated apoA-I, at 24 h incubation, exhibited a 20% lower FRA than apoA-I alone, but the co-addition of CIGB-258 (final 100 μM) caused a significant increase in FRA, up to 27% in a dose-dependent manner during 24 h incubation. At 48 h incubation, the same increased tendency in FRA was detected in a mixture of apoA-I plus CIGB-258, even though the FRA was not elevated in a time-dependent manner. Interestingly, CIGB-258 (final 100 μM) alone did not display an FRA of approximately 25% of the apoA-I activity, but a mixture of apoA-I and CIGB-258 (final 100 μM) exerted an up to 155% increase in FRA. These results suggest that there might be a synergistic interaction between CIGB-258 and apoA-I to the increase in FRA via the enhancement of apoA-I stabilization

### 2.4. Production of IL-6 by Microinjection of CML

Twenty-four hours post-injection, as shown in Figure 4, the CML-alone injected embryo (photo b) showed the highest IL-6 area, 7.8-fold higher than the PBS-alone injected embryo (photo a). In contrast, co-injection of apoA-I or CIGB-258 caused a 50% and 85% reduction in the IL-6 stained area, respectively. Survivability was decreased remarkably up to 21% in the CML-alone injected embryos, while the PBS-injected group showed 78% survivability. In the presence of CML, co-injection of apoA-I or CIGB-258 resulted in 46% and 57% survivability, respectively. These results showed that microinjection of CML into zebrafish embryos caused acute death that was closely associated with IL-6 production, suggesting that the CML injection caused proinflammatory death with IL-6 production. Furthermore, embryo death with high inflammation by IL-6 production was ameliorated by apoA-I or CIGB-258 co-injection.

### 2.5. Embryo Development and Survivability

As shown in Figure 5A, microinjection of CML into zebrafish embryos resulted in only 21% survivability during 24 h post-injection, while the PBS-injected embryo showed 80% survivability. In the presence of CML, a co-injection of either Infliximab or Etanercept resulted in 40–44% survivability, while a co-injection of Tocilizumab or CIGB-258 showed higher survivability of approximately 51 ± 2% and 47 ± 4%, respectively. At 5 h post-injection, all drug-injected embryos showed similar survivability (61–62%) regardless of the drug species in the presence of CML, while the CML-alone injected zebrafish showed the lowest survivability, ~43 ± 6%.

The CML-alone injected embryo showed developmental defects and early death, as indicated by the red arrowhead at 5 h post-injection, while the drug-co-injected embryo showed normal developmental morphology and survivability (Figure 5B). At 24 h post-injection, the PBS-injected embryo showed a normal developmental speed and embryo morphology at the primordium-6 stage with eye pigmentation and tail elongation with more than 32 somites. Conversely, CML injected embryo remained at the 21-somite stage with developmental defects in the eye and tail, as indicated by the red arrowhead, but a co-injection of CIGB-258 recovered the normal development speed and morphology.

DHE staining showed that the CML-alone injected embryo (photograph b) had the strongest red intensity. In contrast, the PBS-injected embryo (photograph a) showed the weakest red intensity, as shown in Figure 5B,C. Among the drug-injected embryos, the Tocilizumab-injected embryo (photograph e) and CIGB-258-injected embryo (photograph f) showed a weaker red intensity than the Infliximab-injected (photograph c) and Etanercept-injected embryos (photograph d). Acridine Orange (AO) staining to detect the extent of apoptosis showed that the CML-alone injected embryo exhibited the strongest green fluorescence intensity, suggesting that the CML induced the highest apoptosis, as shown in Figure 5B,C. Interestingly, the fluorescence-merged image showed that the CML-injected site was stained by DHE and acridine orange, indicating simultaneous ROS production and apoptosis in the embryo (photograph b, Figure 5B). Meanwhile, a co-injection of Tocilizumab and CIGB-258 showed the weakest fluorescence, indicating that the drugs inhibited cellular apoptosis. In contrast, the Infliximab and Etanercept groups showed higher fluorescence than the Tocilizumab and CIGB-258 groups. Infliximab and Etanercept shared a similar extent in ROS production and apoptosis, with approximately 44–45% survivability. These results showed that the IL-6 inhibitor displayed higher anti-inflammatory activity than the TNF-α inhibitors in zebrafish embryos.

### 2.6. Recovery from Acute Paralysis of Hyperlipidemic Zebrafish

Hyperlipidemic zebrafish were prepared by one month of feeding with a 4% cholesterol diet before the intraperitoneal injection of CML and CIGB-258. As a preliminary test, the hyperlipidemic zebrafish were more resistant to acute paralysis induced by CML (final 250 μg, approximately 3 mM in 300 mg of zebrafish body weight). After the preliminary test with 250 μg (3 mM), 375 μg (final 4.5 mM), and 500 μg (final 6 mM) of CML in the hyperlipidemic zebrafish, 500 μg of CML injection caused acute paralysis until 20 min post-injection. As shown in photograph a, Figure 6A, all fish were lying down on the bottom of the cage with occasional trembling without movement or swimming at 20 min post-injection. Although the CML-injected zebrafish could not swim in the bottom of the cage, they were still alive but shuddering, as shown in Appendix A. At 60 min post-injection (photograph b), 20 ± 5% of zebrafish could swim again with 23 ± 6% survivability, but the swimming pattern showed staggering and irregular movements. At 180 min, only 7 ± 3% of zebrafish in the CML-alone group could swim again with 13 ± 8% survivability.

A co-injection of CIGB-258 (final 1 μg in PBS) and CML (final 6 mM) accelerated the recovery of the swimming ability from paralysis with higher survivability (Figure 6B,C). At 20 min post-injection, 10 ± 5% of the zebrafish in the CIGB-258 group could swim with 83 ± 3% survivability, as shown in Appendix A, while no fish could swim in the CML-alone group. At 60 min post-injection, 43 ± 3% of zebrafish in the CIGB-258 group could swim again with 57 ± 6% survivability. At 180 min, 50 ± 1% of zebrafish in the CIGB-258 group could swim again with 57 ± 1% survivability.

The first death of the fish appeared at 6 ± 5 min and 13 ± 5 min post-injection in the CML + PBS group and CML + CIGB-258 group, respectively, suggesting that the CIGB-258 treatment induced two-fold slower neurotoxicity and protected the hyperlipidemic zebrafish from acute death caused by the CML injection. At 20 min and 60 min post-injection, the CIGB-258 group showed 1.3-fold (*p* = 0.013) and 2.4-fold (*p* = 0.024) higher survivability, respectively, than the CML-alone group. At 180 min post-injection, the CML-alone group showed 13 ± 8% survivability, while the CIGB-258 group showed 57 ± 1% survivability (*p* = 0.010, Figure 6C). These results suggest that the co-presence of CIGB-258 could induce faster recovery and survivability up to 4.2-fold higher than the CML alone group by ameliorating acute paralysis and death by the toxicity of CML in adult zebrafish.

### 2.7. Histology Analysis

Histology analysis of the liver with Hematoxylin and Eosin (H&E) staining revealed the HCD-alone group (photograph b) to have a 1.3-fold more H &E-stained area than the ND-alone group (photograph a), as shown in Figure 7A, suggesting that HCD supplementation for 30 days caused more infiltration of neutrophils with an inflammatory process. The HCD + CML group (photograph c) showed a 1.5-fold stronger stained area of the nucleus than the HCD-alone group (photograph b), as shown in Figure 7A. Meanwhile, a co-injection of CIGB-258 caused a 39% decrease in the H&E-stained area. Quantification analysis revealed 1.6 times more neutrophil infiltration in the CML group than in the CIGB-258 group, indicating that hepatic inflammation was reduced significantly by a co-injection of CIGB-258 (Figure 7B).

As shown in Figure 8A, Oil red O staining showed that the HCD group exhibited the strongest red intensity, 2.7-fold more red intensity than the normal diet (ND) group. Hence, HCD consumption for eight weeks caused remarkable fatty liver changes. Under HCD, the CML + PBS group showed a 3.3-fold higher red intensity than the CML + CIGB-258 group, suggesting that the co-injection of CIGB-258 caused a 70% decrease in fatty liver changes (Figure 8B). DHE staining revealed the strongest red intensity in the CML group; 2.3-fold stronger than in the HCD-alone group. Meanwhile, a co-injection of CIGB-258 ameliorated ROS production, resulting in a 38% decrease in the DHE-stained area.

### 2.8. Comparison of the Anti-Inflammatory Efficacy among TNF-α Inhibitors, IL-6 Inhibitors, and CIGB-258

The efficacy of the anti-inflammatory activity with an equal dosage was compared by injecting approximately 1 μM of CIGB-258 (1 μg), Infliximab (Remsima, final 43 μg), Etanercept (Enbrel, final 15 μg), or Tocilizumab (Actemra, final 44 μg) individually into the zebrafish (approximately 300 mg of body weight) in the presence of an equal amount of CML (500 μg, around final 6 mM in zebrafish body weight). Infliximab and Etanercept (TNF-α inhibitors) and Tocilizumab (IL-6 inhibitor) are currently prescribed to treat rheumatoid arthritis. At 30 min post-injection, the zebrafish in the CML alone group was paralyzed and lying down on the bottom of the cage due to severe neurotoxicity (Figure 9A). Conversely, the CIGB-258 group and Tocilizumab group showed a faster appearance of swimming fish with a more active movement pattern. At 120 min post-injection, the CIGB-258 and Tocilizumab group fully recovered their normal swimming pattern with a stabilized posture. In contrast, the Infliximab and Etanercept groups showed staggered movement with occasional cramping. After 180 min post-injection in hyperlipidemic zebrafish, as shown in Figure 9A,B, the CIGB-258 and Tocilizumab groups showed the highest swimming ability (~60 ± 4%), whereas the Infliximab and Etanercept groups showed significantly poorer swimming ability (30 ± 4% and 20 ± 4%, respectively) (Figure 9B).

Under HCD, the CIGB-258 and Tocilizumab groups showed the highest survivability of approximately 75 ± 3% at 180 min post-injection, while the Infliximab group and Etanercept group showed lower survivability of approximately 55 ± 5% (Figure 9B). These results showed that the IL-6 inhibitor (Tocilizumab) and CIGB-258 were superior in attenuating the acute inflammation of CML than the TNF-α inhibitors (Infliximab and Etanercept), resulting in faster recovery of the swimming ability and higher survivability (Figure 9).

### 2.9. Histology and Immunohistochemistry Analyses

Histology analysis with H&E staining showed that the CML-alone group (PBS, photograph b) with HCD showed the most intense red-stained area (Figure 10A), 1.7-fold higher than the PBS alone group, indicating the number of infiltrated neutrophils. In contrast, the CIGB-258 group (photograph f) showed the lowest number of neutrophils, ~29% lower than the CML-alone group (*p* = 0.031), as shown in Figure 10A. The Etanercept and Tocilizumab groups showed a similar H&E-stained pattern: approximately 4% and 11% lower than the CML-alone group, respectively. The CIGB-258 group showed the smallest H&E-stained area (~26.5 ± 0.8%), while the Infliximab, Etanercept, and Tocilizumab groups showed a 37.4 ± 2.3%, 35.6 ± 1.9%, and 33.1 ± 0.5% stained area, respectively (Figure 10B).

Oil red O staining showed that the CML group (29 ± 5% of the stained area) had 4.1-fold more red intensity than the PBS group (7 ± 1% of the stained area), indicating that the CML injection caused severe fatty liver changes (Figure 11A). Meanwhile, the CIGB-258 and Tocilizumab groups showed the least stained area (~8 ± 2% and 10 ± 2%, respectively), indicating that a co-injection of CIGB-258 ameliorated the fatty liver changes with the highest efficacy. The Infliximab and Etanercept groups (TNF-α inhibitors) showed similar efficacy in reducing the fatty liver changes (~18–20% of the stained area, Figure 11B).

DHE staining also showed that the CML-alone group had the strongest red intensity (~32 ± 2% stained area), which was 2.5-fold higher than that of the PBS-alone group (~13 ± 1% stained area). Among the drugs, the CIGB-258 group showed the weakest DHE-stained area (~18 ± 3% of the stained area), whereas the Infliximab group showed the highest intensity (29 ± 2%). The Tocilizumab and Etanercept groups showed similar ROS production (~25–26% stained area), which was higher than the CIGB-258 group.

As shown in Figure 12A, immunohistochemistry detection of IL-6 with hepatic tissue showed that the CML-alone group showed the largest stained area (~29 ± 2%), which was 4.8 times higher than that of the PBS-alone group. Hence, the CML injection caused severe IL-6 accumulation in hepatic tissue. Meanwhile, the CIGB-258 group showed the least IL-6 stained area (~10 ± 1%), whereas the Tocilizumab group showed the second-least IL-6 stained area (~14 ± 3%). The Infliximab and Etanercept groups (TNF-α inhibitors) showed 27 ± 2% and 21 ± 4%, respectively, indicating weaker efficacy than CIGB-258 (Figure 12B).

### 2.10. Blood Lipid Profiles

After 180 min post-injection, all zebrafish were sacrificed for blood collection in the presence of ethylenediaminetetraacetic acid (EDTA) as an anticoagulant. As shown in Figure 13, the serum lipid profiles of the HCD-fed zebrafish showed that the CML-alone group had the highest plasma total cholesterol (TC) and TG levels of approximately 372 ± 22 mg/dL and 378 ± 34 mg/dL, respectively. In contrast, the CIGB-258 group showed the lowest TC and TG levels of approximately 202 ± 10 mg/dL and 207 ± 15 mg/dL, respectively. The Infliximab and Etanercept groups showed a similar plasma lipid level to the CML-alone group (~305 ± 16 mg/dL and 319 ± 24 mg/dL of TC, respectively, and 357 ± 22 mg/dL and 362 ± 17 mg/dL of TG, respectively). The Tocilizumab group showed slightly lower plasma TC levels and higher TG levels than the CIGB-258 group (~173 ± 11 mg/dL and 222 ± 25 mg/dL of TC and TG, respectively). Hence, the CIGB-258 treatment caused a 46% decrease in the plasma TC and TG levels compared to the CML-alone group.

## 3. Discussion

Acute inflammation and paralysis were induced by a CML injection in hyperlipidemic zebrafish with hepatic inflammation and increased proinflammatory cytokine (IL-6) and blood TG levels via putative neurotoxicity. This study examined the protective effect of CIGB-258 for HDL/apoA-I against glycation stress and its anti-inflammatory activity in a hyperlipidemic and hyperinflammatory zebrafish model. Treatment of CML in HDL_3_ caused the production of AGE with proteolytic degradation of apoA-I (Figure 1). CIGB-258 protected apoA-I from CML-induced glycation and proteolytic degradation (Figure 2) by stabilizing the apoA-I secondary structure (Figure 3). The Trp fluorescence of the apoA-I and CIGB-258 mixture would have originated from Trp108 of apoA-I because there is no Trp in the amino acid sequence of CIGB-258. The CML-treated apoA-I showed a 3.1 nm red-shift in Trp WMF with increasing yellowish fluorescence intensity and proteolytic degradation. These results agree with the previous report that fructated apoA-I exhibited a 5 nm red-shift in WMF [35] with multimerization and a loss of antioxidant ability [36]. A fructose treatment of apoA-I caused glycation to produce an AGE product with yellowish fluorescence and the exposure of Trp to the aqueous phase [36,37]. The results showed that the CML treatment, a major AGE, also causes the production of yellowish fluorescence and exposure of Trp in apoA-I toward the aqueous phase, similar to a fructose treatment.

Microinjection of CML into zebrafish embryos caused acute inflammatory death with severe IL-6 production (Figure 4), suggesting that the presence of CML might play a role in the inflammatory cascade. Co-injection of apoA-I caused a 50% decrease in IL-6 production, showing a good agreement with previous reports that apoA-I is an anti-inflammatory protein [38,39]. Furthermore, co-injection of CIGB-258 could block proinflammatory signaling to produce IL-6 and induce higher survivability, suggesting that CIGB-258 has more potent anti-inflammatory activity than lipid-free apoA-I.

A recent report suggested that the CML can undergo a second glycation event at its secondary amino group, leading to a new class of Amadori rearrangement products [40]. Glycated amino acids, such as CML, are not inert molecules because CML can be degraded and converted into secondary amino groups [41], which can be subject to second glycation events. These reports and the current study suggest that HDL, a macromolecule complex with lipids and proteins, can be modified via a putative interaction with the CML, as shown in Figure 1 and Figure 2, such as glycoxidation or lipid peroxidation. Further study is necessary to investigate a more precise molecular interaction mechanism between the CML and HDL/apoA-I to induce the modification and production of dysfunctional HDL.

The red-shift in fluorescence is associated with the exposure of Trp to the aqueous phase, suggesting the partial collapse of the secondary structure of apoA-I through the glycation attack of CML. Meanwhile, adding CIGB-258 restored the secondary structure through a 3 nm blue-shift of Trp in apoA-I, suggesting that the movement of Trp toward the nonpolar phase stabilizes the secondary structure. These results suggest that there might be a putative synergistic interaction between apoA-I and CIGB-258 to maximize structural stability. These stabilization data are in good agreement with a previous study [21] showing that CIGB-258 prevented the degradation of HDL, displaying antioxidant activity against a CML or ferrous ion treatment. The protection of apoA-I (Figure 2 and Figure 3A) might be linked to the enhanced antioxidant activity in vitro (Figure 3B), protection of acute embryo death from the inflammation of CML (Figure 4 and Figure 5), and amelioration of the acute inflammatory response (Figure 6, Figure 7, Figure 8, Figure 9, Figure 10, Figure 11 and Figure 12) with lower blood lipids in adult zebrafish (Figure 13).

Microinjections of CML (500 ng) into zebrafish embryos also resulted in the death of embryos with a production of IL-6 (Figure 4) and severe developmental defects (Figure 4 and Figure 5). Interestingly, the CML-alone-injected site, as indicated by the arrow, showed the highest ROS production and apoptosis (photograph b, Figure 5B), indicating the simultaneous occurrence of oxidative damage and cell death. IL-6 inhibitor (Tocilizumab)-injected and CIGB-258-injected embryo exhibited less ROS production and apoptosis than the TNF-α inhibitors (Infliximab and Etanercept), but the four drugs reduced ROS production and apoptosis. An injection of CML caused IL-6 production (Figure 4), and ROS production (Figure 5) in the embryo via the inflammatory cascade process. In contrast, the co-presence of an IL-6 inhibitor and TNF-α inhibitor alleviated ROS production in embryos (Figure 5). In addition, the injection of an IL-6 inhibitor (Tocilizumab) showed better embryo development and survivability than that of a TNF-α inhibitor (Infliximab and Etanercept) to inhibit CML-induced inflammation.

An intraperitoneal injection of CML into adult hyperlipidemic zebrafish resulted in severe inflammatory damage in hepatic tissue: more infiltration of neutrophils (Figure 7 and Figure 9), fatty liver changes (Figure 8A), ROS production (Figure 8B and Figure 11B), and IL-6 expression (Figure 12). These results showed good agreement with a previous report with normolipidemic zebrafish [21], indicating that the CIGB-258 group had the least infiltration of neutrophils in hepatic tissue. A co-injection of monoclonal antibodies or CIGB-258 also caused the same efficacy tendency in ameliorating the acute inflammatory response in adult zebrafish, as shown in Figure 9, Figure 10, Figure 11, Figure 12 and Figure 13. The CIGB-258 and Tocilizumab groups showed a similar decrease in plasma TC and TG levels, approximately 46% lower than the CML-alone group. Conversely, the Infliximab treatment caused a 19% and 6% decrease in plasma TC and TG levels, respectively, even though the Etanercept group showed only a 15% decrease in TC without a decrease in TG compared to the CML-alone group. These results suggest that IL-6 inhibitors, Tocilizumab and CIGB-258, produce similar improvements in the plasma lipid profiles, while TNF-α inhibitors (Infliximab and Etanercept) did not improve the lipid profile and hepatic inflammation caused by CML treatment in zebrafish. A previous study reported that CIGB-258 caused a 32% and 43% decrease in serum TC and TG levels in normolipidemic zebrafish, respectively. Similar decreases in the blood lipid profile and hepatic inflammation were observed in the CIGB-258 group of both normolipidemic zebrafish and hyperlipidemic zebrafish.

A high serum level of IL-6 has been linked with dyslipidemia, as shown in patients with oral lichen planus (OLP), a chronic inflammatory disease affecting the mucus membrane of the oral cavity [42]. The OLP group showed higher serum IL-6, TG, and cholesterol levels than the control groups with more atherogenic risk factors. Positive correlations exist between the serum levels of TG and IL-6 with weight regain and fat mass expansion [43]. The decrease in blood TG is directly associated with ameliorating the inflammatory response in the acute phase because the intravenous administration of IL-6 stimulated the highest level of TG secretion in the blood at 2 h post-injection in a dose-dependent manner [44]. Moreover, patients with psoriatic arthritis showed higher serum IL-6, TC, and LDL-C levels than patients with psoriasis alone [45], indicating that IL-6 might be related to dyslipidemia in patients with autoimmune disease. Overall, previous studies and the current results agree with respect to the association of elevated IL-6 levels with increased blood TC and TG levels.

At the same time, the InChianti study reported that low HDL-C is associated with high plasma IL-6 (*r* = −0.23, *p* < 0.01) and TG (*r* = −0.44, *p* < 0.01) levels in a cohort of older adults, 65–102 years old [46], indicating that HDL-C is inversely correlated with the inflammatory parameters. Furthermore, human plasma HDLs and reconstituted HDLs inhibited IL-6 production in human endothelial cells in a concentration-dependent manner by inhibiting p38 MAP kinase [47]. HDL downregulated the IL-6 mRNA levels, and subjects with low HDL-C showed significantly elevated plasma IL-6 levels [47,48]. IL-6 upregulated the synthesis of acute phase proteins in hepatocytes, such as high sensitive C reactive protein (hsCRP), and inhibited regulatory T (T_reg_) cells [49]. Therefore, enhanced HDL stability by CIGB-258 inhibits IL-6 production (Figure 12). The lower IL-6 expression (Figure 4 and Figure 12) means that T_reg_ cells could be stimulated further to exert anti-inflammatory activity in CML-injected zebrafish embryos (Figure 5) and adults (Figure 6). These results are in good agreement with previous reports showing an anti-inflammatory effect of CIGB-258 associated with increased T_reg_ activity in preclinical models of rheumatoid arthritis [33,34].

In terms of a comparison between this paper and our previous paper [21], there are distinct differences between the two; in the current study: (1) the target protein for protection against CML toxicity is lipid-free apoA-I; (2) there is a comparison of embryo survivability among monoclonal antibodies, Infliximab, Etanercept, and Tocilizumab; (3) there is a comparison of adult zebrafish survivability in a hyperlipidemic state with a higher dose of CML (final 6 mM). The current study shows that CIGB-258 protected lipid-free apoA-I (Figure 2 and Figure 3) against CML toxicity from proteolytic degradation, while our previous paper showed that CIGB-258 protected only HDL, not lipid-free apoA-I. The current study revealed that microinjection of CML caused IL-6 production with acute death in the zebrafish embryo (Figure 4). The acute death of the embryo was prevented by co-injection of CIGB-258 and monoclonal antibodies (Figure 5). Acute neurotoxicity and paralysis in hyperlipidemic zebrafish by CML administration (final 6 mM) were also protected by the co-presence of CIGB-258 via inhibition of hepatic inflammation and IL-6 production (Figure 6, Figure 7, Figure 8, Figure 9, Figure 10, Figure 11, Figure 12 and Figure 13).

In conclusion, CIGB-258 treatment improved the structural stabilization of lipid-free apoA-I against the glycation stress of CML to maintain its antioxidant activity. Stabilizing apoA-I suppressed inflammation and improved the survivability of embryos and hyperlipidemic zebrafish by protecting them from CML-injected toxicity and anti-inflammatory activity. These results suggest that CIGB-258 protected hyperlipidemic zebrafish from CML toxicity, such as acute inflammation, paralysis, and death, via stabilization of apoA-I and HDL.

## 4. Materials and Methods

*N*-ε-carboxylmethyllysine (CML, CAS-No 941689-36-7, Cat#14580) with at least >97% purity (TLC) with water (<12.0%), dihydroethidium (DHE, 104821-25-2, Cat #37291), and acridine orange (AO, 65-61-2, Cat#A9231) were purchased from Sigma–Aldrich (St. Louis, MO, USA). CIGB-258 (Jusvinza^®^), a recombinant peptide from HSP60 with 27 amino acids in a lyophilized powder formula (1.25 mg/vial, Lot# 1125J1/0), was obtained from the Center for Genetic Engineering and Biotechnology (CIGB, Havana, Cuba) under the agreement that it would be used for research purposes only. Remsima (Infliximab 100 mg powder, Celltrion, Incheon, South Korea Lot# 2B3C091), Enbrel (Etanercept 50 mg, Pfizer, Dublin, Ireland), and Actemra (Tocilizumab 400 mg/20 mL, Lot# 21K010E11778, JW Pharmaceutical, Seoul, Republic of Korea) were purchased from Shinsung Pharmaceutical (Seoul, Republic of Korea) under the agreement that it would be used for research purposes only.

### 4.1. Purification of Lipoproteins

LDL (1.019 < d < 1.063), HDL_2_ (1.063 < d < 1.125), and HDL_3_ (1.125 < d < 1.225) were isolated from the sera of young and healthy human males (mean age, 22 ± 2 years), who donated blood voluntarily after fasting overnight by sequential ultracentrifugation. The density was adjusted appropriately by adding NaCl and NaBr, as detailed elsewhere [50], and the procedures were carried out according to the standard protocols [51]. The samples were centrifuged for 24 h at 10 °C at 100,000× *g* using a Himac CP100-NX with a fixed angle rotor P50AT4 (Hitachi, Tokyo, Japan) at the Raydel Research Institute (Daegu, Republic of Korea). After centrifugation, each lipoprotein sample was dialyzed extensively against Tris-buffered saline (TBS; 10 mM Tris-HCl, 140 mM NaCl, and 5 mM EDTA [pH 8.0]) for 24 h to remove NaBr.

### 4.2. Purification of apoA-I

ApoA-I was purified from the HDL, which was isolated from human plasma using ultracentrifugation using column chromatography and organic solvent extraction following the method described by Brewer et al. [52]. The purified apoA-I was lyophilized and stored at −80 °C until use.

### 4.3. Glycation of HDL and apoA-I with CML

Glycation was conducted by incubating the purified HDL_3_ (2 mg/mL) with CML (final 100~400 μM) at 37 °C in an atmosphere containing 5% CO_2_ for up to 144 h in 200 mM potassium phosphate/0.02% sodium azide buffer (pH 7.4). The anti-glycation activity was tested by incubating lipid-free apoA-I (final 2 mg/mL) in the same buffer with CML (final 200 μM) in the presence and absence of CIGB-258.

The apoA-I content was compared using SDS-PAGE (5 μg of apoA-I in a lipid-bound state or 7 μg of apoA-I in a lipid-free state per lane) and densitometric analysis because the glycation resulted in a severe decrease in the protein content in HDL and several beneficial functions of apoA-I [53,54]. The extent of the advanced glycation reactions was determined by reading the fluorescence intensities at 370 nm (excitation) and 440 nm (emission), as described previously [55]. The BI was compared by band scanning with Chemi-Doc^®^ XR (Bio-Rad) using Quantity One software (version 4.5.2) from three independent SDS-PAGE analyses.

### 4.4. Wavelength Maximum Fluorescence of HDL

The change in secondary structure upon treatment with ferrous ions was observed at the WMF of the tryptophan residues in HDL_3_. The WMF was determined from the uncorrected spectra obtained on an FL6500 spectrofluorometer (Perkin–Elmer, Norwalk, CT, USA) using Spectrum FL software version 1.2.0.583 (Perkin–Elmer) and a 1 cm path-length Suprasil quartz cuvette (Fisher Scientific, Pittsburgh, PA, USA). The samples were excited at 295 nm to avoid tyrosine fluorescence. As described previously, the emission spectra were scanned from 305 to 400 nm at room temperature [56].

### 4.5. Ferric Ion Reduction Ability

The ferric ion-reducing ability (FRA) was determined using the method reported by Benzie and Strain [57]. Briefly, the FRA reagents were freshly prepared by mixing 20 mL of 0.2 M acetate buffer (pH 3.6), 2.5 mL of 10 mM 2,4,6-tripyridyl-S-triazine (Fluka Chemicals, Buchs, Switzerland), and 2.5 mL of 20 mM FeCl_3_∙6H_2_O. The antioxidant activities of CIGB-258 and apoA-I were estimated by measuring the increased absorbance induced by the ferrous ions. Freshly prepared FRA reagent (300 μL) was mixed with CIGB-258 and apoA-I as an antioxidant source. The FRA was then determined by measuring the absorbance at 593 nm every two minutes during a 60 min period at 25 °C using a UV-2600i spectrophotometer.

### 4.6. Zebrafish

Zebrafish and embryos were maintained using the standard protocols [58] according to the *Guide for the Care and Use of Laboratory Animals* [59]. The maintenance of the zebrafish and procedures using zebrafish were approved by the Committee of Animal Care and the Use of Raydel Research Institute (approval code RRI-20-003, Daegu, Republic of Korea). The fish were maintained in a system cage at 28 °C during treatment under a 10:14 h light cycle with the consumption of normal tetrabit (TetrabitGmbh D49304, 47.5% crude protein, 6.5% crude fat, 2.0% crude fiber, 10.5% crude ash, containing vitamin A [29770 IU/kg], vitamin D3 [1860 IU/kg], vitamin E [200 mg/kg], and vitamin C [137 mg/kg]; Melle, Germany).

### 4.7. Microinjection of Zebrafish Embryos

Embryos at one-day post-fertilization (dpf) were injected individually by microinjection using a pneumatic picopump (PV830; World Precision Instruments, Sarasota, FL, USA) equipped with a magnetic manipulator (MM33; Kantec, Bensenville, IL, USA) with a pulled microcapillary pipette-using device (PC-10; Narishigen, Tokyo, Japan). The injections were performed at the same position on the yolk to minimize bias, as described elsewhere [60]. CML (500 ng) alone, CIGB-258 (1 ng), Infliximab (43 ng), Etanercept (15 ng), or Tocilizumab (44 ng) were injected into the flasks of embryos (final volume 5 nL). After the injection, the live embryos were observed under a stereomicroscope (Motic SMZ 168; Hong Kong) and photographed (20× magnification) using a Motic cam2300 CCD camera. The survivability of the embryo was assessed using the OECD guidelines 2013 [61] with coagulation of the embryo, non-detachment of the tail, lack of somite, and lack of heartbeat. At 24 h post-injection, each live embryo was compared after removing the chorion to compare the developmental stage at higher magnification (50×).

### 4.8. Imaging of Oxidative Stress, Apoptosis, and IL-6 in Embryo

After injecting CML with each drug, the reactive oxygen species (ROS) levels in the embryos were imaged by dihydroethidium (DHE) staining, as described previously [62]. The images of the ROS were obtained by fluorescence observations (Ex = 585 nm and Em = 615 nm). The extent of cellular apoptosis among the groups was compared by acridine orange (AO) staining and fluorescence observations (Ex = 505 nm, Em = 535 nm) using a Nikon Eclipse TE2000 microscope (Tokyo, Japan).

After the injection, the embryos were subjected to immunofluorescence staining to visualize IL-6 production using an anti-IL-6 antibody (ab208113, Abcam London, UK) as primary antibody (1:100 diluted) and goat anti-Rabbit IgG H&L (Alexa Fluor^®^ 488, ab150077, Abcam London, UK) as a secondary antibody (1:300 diluted). The embryos were fixed with para-formaldehyde at 24 hpf and immunohistochemical staining with the primary antibody for 24 h, according to a previous report [63]. The fluorescence area of IL-6 was calculated using Image J software version 1.53r (http://rsb.info.nih.gov/ij/ (accessed on 16 January 2023)) to convert the green intensity from the immunostaining.

### 4.9. Intraperitoneal Injection of CML and CIGB-258 into Adult Zebrafish

Acute inflammation was caused by CML administration (final 250 μg in 10 μL of PBS), corresponding to approximately 833 mg/kg, by an intraperitoneal injection using a 28 gauge needle into the abdomen region of zebrafish (approximately 300 mg of body weight), which were anesthetized by submersion in 2-phenoxyethanol (Sigma P1126; St. Louis, MO, USA) in system water (1:1000 dilution). The single injection dosage (833 mg/kg of body weight) in this study was similar to the oral gavage administration dosage of 200–1000 mg/kg of body weight in a mouse model, as reported previously [23].

Up to 180 min post-injection, the swimming ability and survivability in the CML-alone group and CIGB-258-co-injected group were compared because the CML treatment caused acute paralysis, with lying down on the bottom of the cage without any swimming ability in the preliminary test. The swimming ability was evaluated by the tail fin movement and lack of body convulsion or cramping for 10 s after being paralyzed by the CML injection. The paralyzed animals display short, frequent bursts by overusing their pectoral fins [64]. The death of the zebrafish was assessed by a combination of criteria, including a loss of balance, head up or down, and floating on the surface or sinking, according to the OECD guidelines 2019 [65]. Fish are considered dead if there is no visible movement including gill movement. The anti-inflammatory effects of CIGB-258 against the neurotoxicity of CML were compared with Infliximab and Tocilizumab, a TNF-α and IL-6 inhibitor, respectively. After 180 min post-injection, all zebrafish were sacrificed, and blood was collected for analysis.

### 4.10. Plasma Analysis

Blood (2 μL) was drawn from the hearts of the adult zebrafish, combined with 3 μL of phosphate-buffered saline (PBS)-ethylenediaminetetraacetic acid (EDTA, final concentration, 1 mM) and collected in EDTA-treated tubes. The total plasma cholesterol (TC) and triglyceride (TG) levels were determined using commercial assay kits (cholesterol, T-CHO, and TGs, Cleantech TS-S; Wako Pure Chemical, Osaka, Japan).

### 4.11. Histopathology Analysis

For the morphological tissue observations using Hematoxylin and Eosin (H & E) staining, some hepatic tissues were fixed with a 10% formaldehyde solution for 24 h. The samples were then exchanged twice with the same solution, dehydrated with double ethanol, and formatted in paraffin, producing a 5 μm thick tissue slice treated with poly-L-lysine. For morphological analysis, the prepared tissue sections were stained with H & E for the liver tissue at ×400 and ×1000 magnification with an optical microscope. The nucleus area was quantified using Image J software version 1.53r (http://rsb.info.nih.gov/ij/ accessed on 15 September 2022) to convert the red intensity from H&E staining.

For oil red O staining, the stained liver samples were then entrenched in Tissue-Tek OCT compound (Thermo, Walldorf, Germany) and frozen. Furthermore, 7 μm sections of these tissues were mounted on 3-aminopropyltriethoxysilane (3-APS)-coated slides and viewed under a Leica microcryotome (model CM1510s, Heidelberg, Germany). Seven successive sectioned slides of each zebrafish were first stained with Oil Red O (Cat # O0625, Sigma, St. Louis, MO, USA) and then counterstained with hematoxylin, which highlighted the fatty streak lesions. The extent of oxidative stress in these tissues was compared by observing the totality of reactive oxygen species (ROS) with dihydroethidium (DHE, cat # 37291; Sigma, St. Louis, MO, USA) using a Nikon Eclipse TE2000 microscope (Tokyo, Japan), as described elsewhere [66]. The section fluorescence was quantified using Image J software version 1.53r (http://rsb.info.nih.gov/ij/ accessed on 16 September 2022).

Immunohistochemistry analysis was carried out with the anti-human IL-6 antibody (ab9324, Abcam, London, UK) as the primary antibody and horseradish peroxidase conjugated-anti mouse immunoglobulin G antibody as the secondary antibody using an Envision+ system (K4001, Dako, Denmark).

### 4.12. Statistical Analysis

The data are expressed as the mean ± SD from at least three independent experiments with duplicate samples. For the in vitro apoA-I study, multiple treatments were compared using a one-way analysis of variance (ANOVA) using a Bonferroni test as a post-hoc analysis. For the in vivo zebrafish study, multiple groups were compared using ANOVA with a Scheffe test. Statistical analysis was performed using the SPSS software program (version 28.0; SPSS, Inc., Chicago, IL, USA). A *p*-value < 0.05 was considered significant.

## Figures and Tables

**Figure 1 ijms-24-07044-f001:**
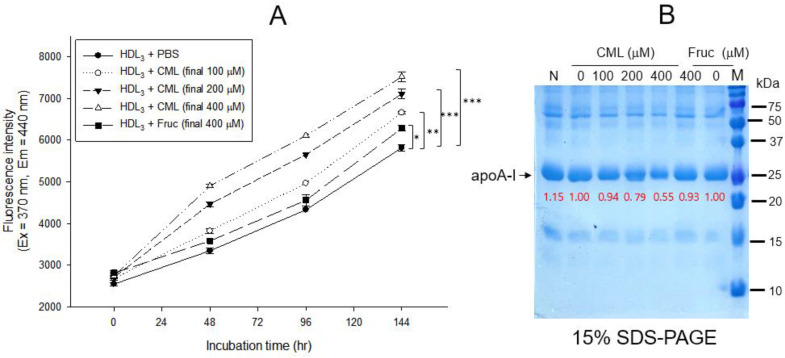
Glycation of HDL_3_ (2 mg/mL) by treatment of carboxymethyllysine (CML) and fructose (fruc) at 37 °C in the presence of 5% CO_2_. (**A**) Comparison of fluorescence intensity (Ex = 370 nm, Em = 440 nm) from the glycated product in HDL_3_ (200 μg) with the treatment of either CML (final 0, 100, 200, and 400 μM) or fructose (final 0 and 400 μM) during 144 h The data are expressed as mean ± SEM from three independent measurements with duplicate samples. *, *p* < 0.05; **, *p* < 0.01; ***, *p* < 0.001. (**B**) Electrophoretic patterns (15% SDS-PAGE) of glycated HDL_3_ with the treatment of either CML (final 0, 100, 200, and 400 μM) or fructose (final 0 and 400 μM) at 144 h. Five micrograms of total protein in HDL_3_ was equally applied to each lane. Red font indicates band intensity from three independent SDS-PAGE analyses. Lane N, native HDL_3_; lane M, molecular weight marker (Bio-Rad, prestained low-range).

**Figure 2 ijms-24-07044-f002:**
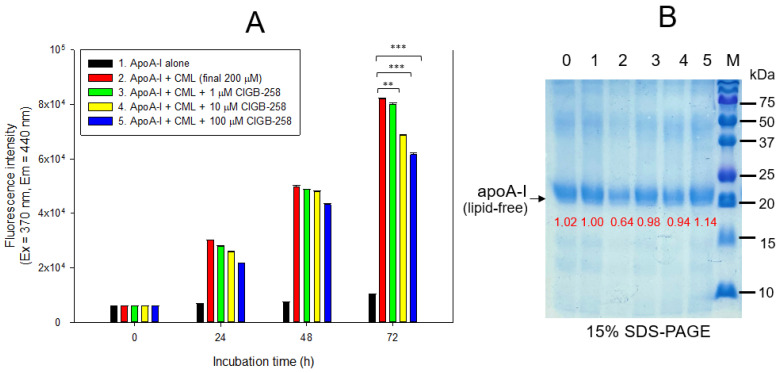
Glycation extent of apoA-I in the lipid-free state by carboxymethyllysine (CML) in the presence and absence of CIGB-258. (**A**) Measurement of the fluorescence intensity (Ex = 370 nm, Em = 440 nm) of apoA-I during 72 h incubation. The data are expressed as mean ± SEM from three independent measurements with duplicate samples. **, *p* < 0.01; ***, *p* < 0.001. (**B**) Electrophoretic patterns of the apoA-I after 72 h incubation (15% SDS-PAGE). Seven μg of protein in apoA-I was equally applied to each lane. Lane 0, apoA-I alone; lane 1, apoA-I (1 mg/mL) + PBS; lane 2, apoA-I + CML (final 200 μM); lane 3, apoA-I + CML + CIGB-258 (final 1 μM); lane 4, apoA-I + CML + CIGB-258 (final 10 μM); lane 5, apoA-I + CML + CIGB-258 (final 100 μM); lane M, molecular weight marker (Bio-Rad, pre-stained low-range).

**Figure 3 ijms-24-07044-f003:**
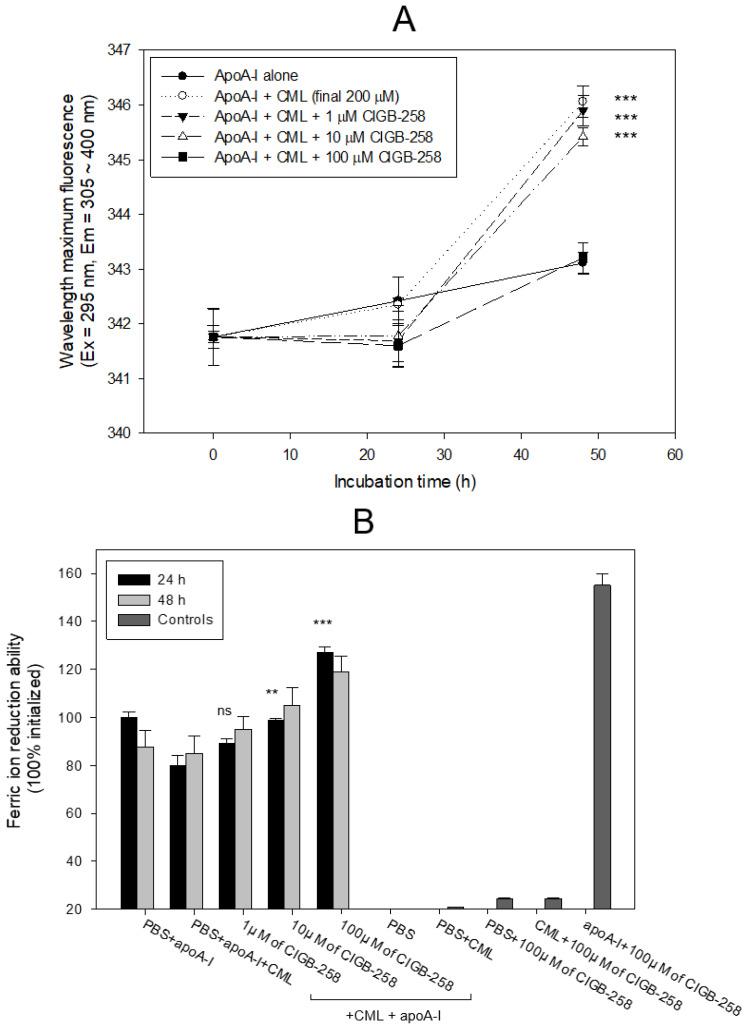
Comparison of apoA-I structure and antioxidant ability in the presence of carboxymethyllysine (CML) and CIGB-258. (**A**) Measurement of the wavelength maximum fluorescence of the Trp emission scanning spectrum (Ex = 295 nm, Em = 305–400 nm) during 48 h incubation with apoA-I and CML under the absence or presence of CIGB-258. The data are expressed as mean ± SEM from three independent measurements with duplicate samples. ***, *p* < 0.001 versus apoA-I alone. (**B**) Measurement of ferric ion reduction ability (FRA) with apoA-I during 60 min incubation with CML and CIGB-258. **, *p* < 0.01 versus PBS + apoA-I + CML; ***, *p* < 0.001 versus PBS + apoA-I + CML; ns, not significant.

**Figure 4 ijms-24-07044-f004:**
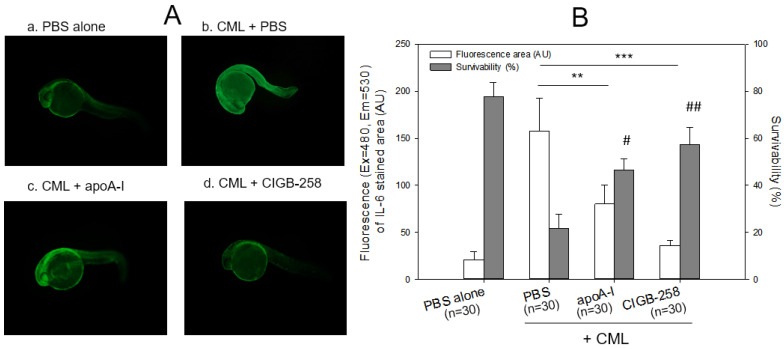
Immunostaining of interleukin (IL)-6 with zebrafish embryo and survivability after microinjection of apoA-I (28 ng) or CIGB-258 (1 ng) with CML (500 ng) and into zebrafish embryo at 1 h post-fertilization. (**A**) Immunofluorescence image of immunostained with anti-IL-6 antibody (ab208113, Abcam London, UK) as primary antibody (1:100 diluted) and goat anti-Rabbit IgG H&L (Alexa Fluor^®^ 488, ab150077, Abcam, London, UK) as a secondary antibody (1:300 diluted) at 24 h post-injection. a. PBS alone; b. CML + PBS; c. CML + apoA-I; d. CML + CIGB-258. (**B**) Calculation of the immunostained fluorescence area (left axis) using Image J software version 1.53r (http://rsb.info.nih.gov/ij/ accessed on 16 January 2023) to convert the green intensity from the immunostaining. Survivability of the embryo at 24 h post-injection was also presented as the right axis. **, *p* < 0.01; ***, *p* < 0.001 of the immunofluorescence. #, *p* < 0.05 versus PBS + CML; ##, *p* < 0.01 versus PBS + CML of the survivability.

**Figure 5 ijms-24-07044-f005:**
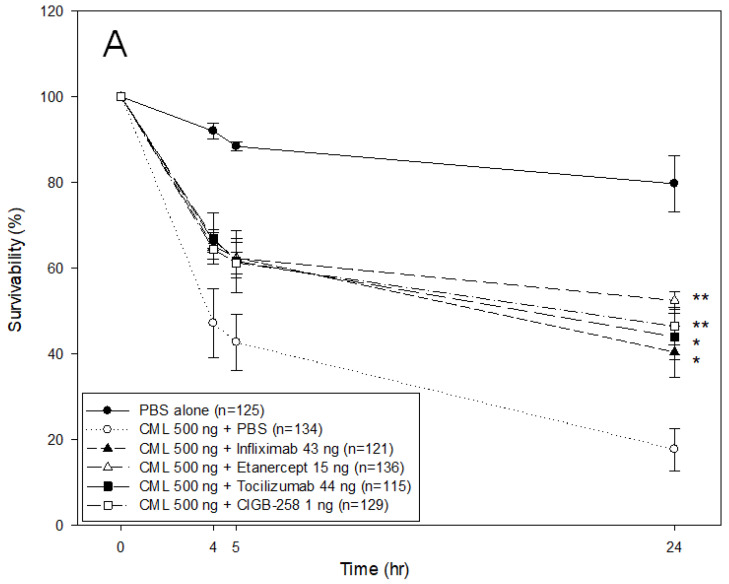
Comparison of the protective activity of monoclonal antibodies and CIGB-258 against carboxymethyllysine (CML, final 500 ng) toxicity in zebrafish embryos. (**A**) Survivability of zebrafish embryos during 24 h post-injection in the presence of Infliximab (43 ng), Etanercept (15 ng), Tocilizumab (44 ng), or CIGB-258 (final 1 ng). *, *p* < 0.05; **, *p* < 0.01. (**B**) Stereo image of the zebrafish embryos at 5 h and 24 h post-injection. The red arrowheads indicate defective development and death of embryos in the CML alone group (photograph b). The blue arrowhead indicates the slowest developmental speed in eye pigmentation and tail elongation in CML alone group at 24 h post-injection (photograph c). Fluorescence image of dihydroethidium (DHE, Ex = 585 nm, Em = 615 nm) stained and acridine orange (AO, Ex = 505 nm, Em = 535 nm) stained embryo at 5 h post-injection. The extent of ROS production and apoptosis was visualized by DHE and AO staining, respectively. The white arrow indicates the CML-injected site to show that ROS production and cellular apoptosis occurred simultaneously by acute inflammation. Photo a, PBS alone; photo b, CML + PBS; photo c, CML + Infliximab; photo d, CML + Etanercept; photo e, CML + Tocilizumab; photo f, CML + CIGB-258 (**C**) Quantification of the fluorescence from DHE-stained and AO-stained embryo images using Image J software version 1.53r (http://rsb.info.nih.gov/ij/ accessed on 17 January 2023). AU, arbitrary unit; NS, not significant.

**Figure 6 ijms-24-07044-f006:**
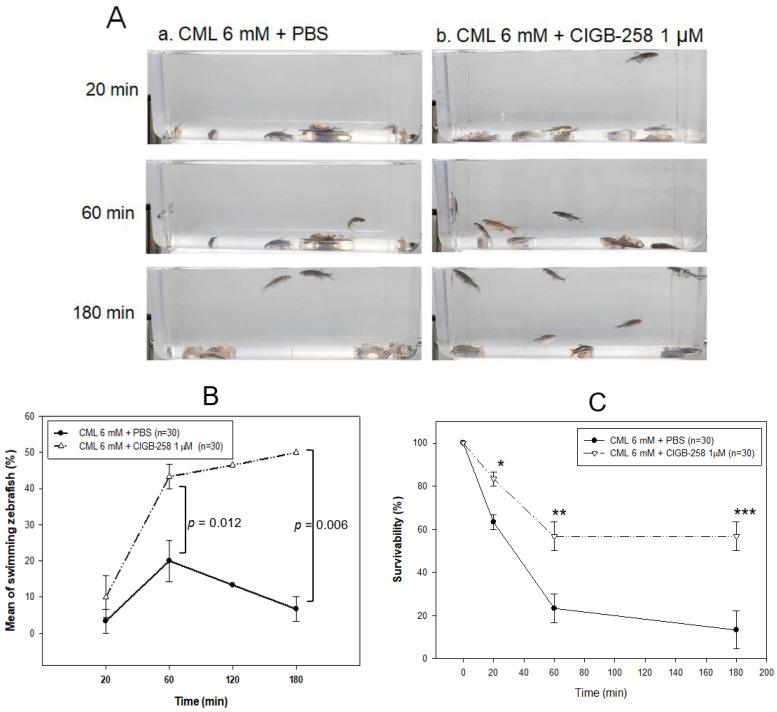
Comparison of the swimming ability after an injection of carboxymethyllysine (CML) with or without CIGB-258. (**A**) Still image of swimming pattern of zebrafish after 20 min, 60 min, and 180 min post-injection of CML (500 μg) + PBS (photograph a) and CML (500 μg) + CIGB-258 (1 μg, photograph b) per fish. (**B**) Percentage of swimming zebrafish after 20 min, 60 min, and 180 min post-injection of CML (500 μg) + PBS and CML (500 μg) + CIGB-258 (1 μg). (**C**) Survivability at 20 min, 60 min, and 180 min post-injection of CML (500 μg) + PBS and CML (500 μg) + CIGB-258 (1 μg). *, *p* < 0.05; **, *p* < 0.01; ***, *p* < 0.001.

**Figure 7 ijms-24-07044-f007:**
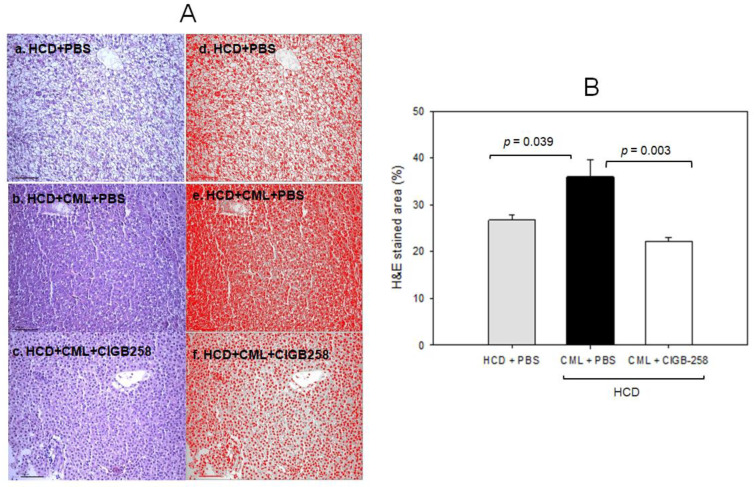
Histologic analysis of hepatic tissue. The zebrafish were injected with carboxymethyllysine (CML) in the presence or absence of CIGB-258 under the consumption of a high-cholesterol diet (HCD). (**A**) Visualization of the infiltration of neutrophils by Hematoxylin & Eosin (H&E) staining as shown in the left photographs (photographs a, b, c). The right photographs show the conversion of the Hematoxylin-stained area into red intensity (photographs d, e, f). (**B**) Quantification graph of the nucleus area from H&E staining using Image J software version 1.53r (http://rsb.info.nih.gov/ij/ (accessed on 17 January 2023)) as shown in red intensity. The cell infiltration area was quantified as a percentage of the total hepatocyte area, as determined in H&E-stained sections. The scale bar indicates 100 μm.

**Figure 8 ijms-24-07044-f008:**
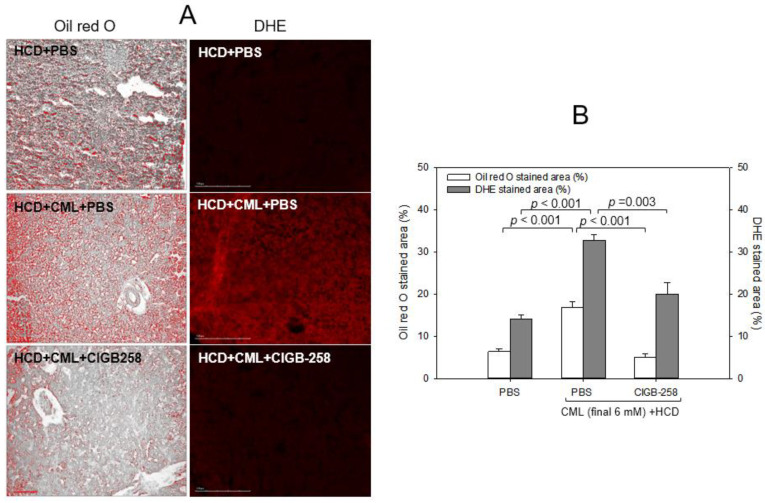
Comparisons of the fatty liver changes and reactive oxygen species (ROS) by Oil red O staining and dihydroethidium (DHE) staining, respectively. (**A**) Representative image of oil red O-stained and DHE-stained hepatic tissue at 180 min post-injection. The scale bar indicates 100 μm. (**B**) Quantification of the oil red O intensity and DHE fluorescence (Ex = 585 nm, Em = 615 nm) intensity using Image J software version 1.53r (http://rsb.info.nih.gov/ij/ accessed on 16 September 2022).

**Figure 9 ijms-24-07044-f009:**
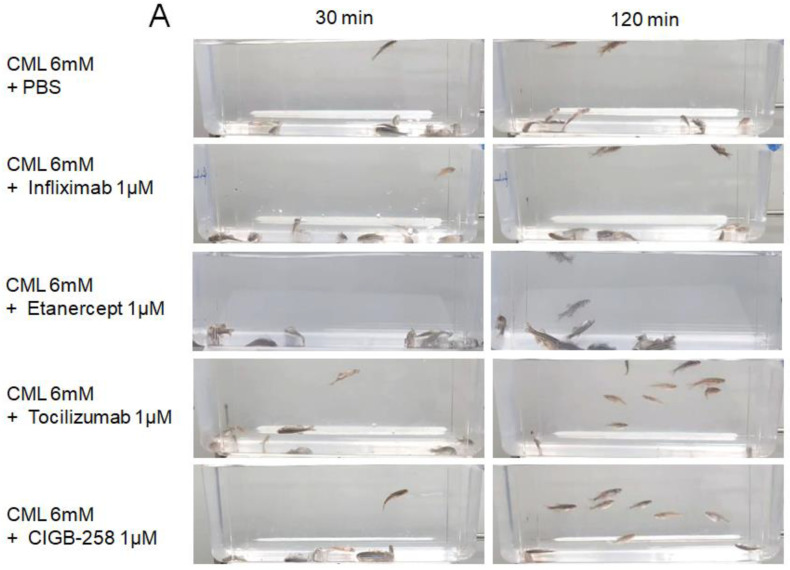
Comparison of the swimming ability and survivability after an intraperitoneal injection of carboxymethyllysine (CML) with Infliximab (Remsima), Etanercept (Enbrel), Tocilizumab (Actemra), or CIGB-258 (Jusvinza). (**A**) Still image of the swimming pattern of zebrafish 30 min and 120 min post-injection of CML with Infliximab (Remsima), Etanercept (Enbrel), Tocilizumab (Actemra), or CIGB-258 (Jusvinza) 30 min and 120 min post-injection. (**B**) Survivability of the zebrafish at 180 min post-injection of CML with Infliximab (Remsima), Etanercept (Enbrel), Tocilizumab (Actemra), or CIGB-258 (Jusvinza). *, *p* < 0.05 versus CML + PBS; ***, *p* < 0.001 versus CML + PBS. (**C**) Percentage of swimming zebrafish after an injection of CML with Infliximab (Remsima), Etanercept (Enbrel), Tocilizumab (Actemra), or CIGB-258 (Jusvinza) at 30 min, 60 min, and 120 min post-injection. *, *p* < 0.05 versus CML + PBS; ***, *p* < 0.001 versus CML + PBS.

**Figure 10 ijms-24-07044-f010:**
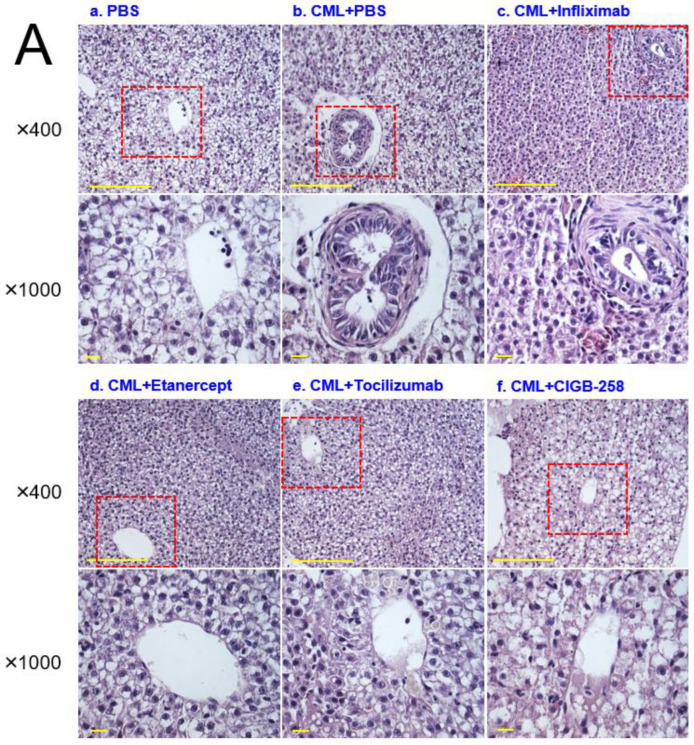
Histologic analysis of the hepatic tissue from zebrafish injected with PBS alone (photograph a), CML+ PBS (photograph b), CML+ Infliximab (photograph c), CML + Etanercept (photograph d), CML + Tocilizumab (photograph e), and CML + CIGB-258 (photograph f) under the consumption of a high cholesterol diet (HCD). (**A**) Photographs show the infiltration of neutrophils by Hematoxylin & Eosin (H&E) staining after converting the Hematoxylin-stained area into a red intensity. The yellow scale bar indicates 100 μm. The red box area in image of 400× magnification was observed again at 1000× magnification (**B**) Quantification of the nucleus area from the H&E staining using Image J software version 1.53r (http://rsb.info.nih.gov/ij/ accessed on 15 September 2022). The statistical significance of the groups was indicated as *p*-values at the top of the graph.

**Figure 11 ijms-24-07044-f011:**
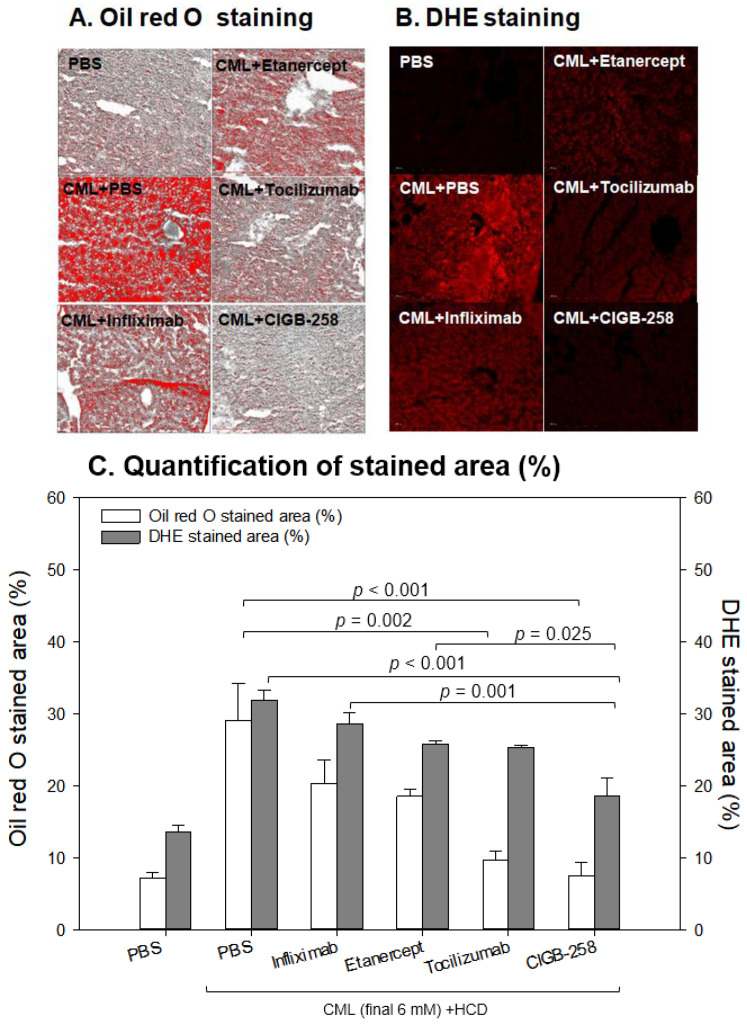
Comparisons of the fatty liver changes (**A**) and production of reactive oxygen species (ROS) (**B**) by Oil red O staining and dihydroethidium (DHE) staining, respectively, among each drug-injected zebrafish. (**A**) Representative image of oil red O-stained hepatic tissue to show fatty liver changes. The scale bar indicates 100 μm. (**B**) Representative fluorescence (Ex = 585 nm, Em = 615 nm) image of DHE-stained hepatic tissue to show ROS production. The scale bar indicates 100 μm. (**C**) Quantification of the oil red O intensity and DHE fluorescence intensity using Image J software version 1.53r (http://rsb.info.nih.gov/ij/ accessed on 15 September 2022).

**Figure 12 ijms-24-07044-f012:**
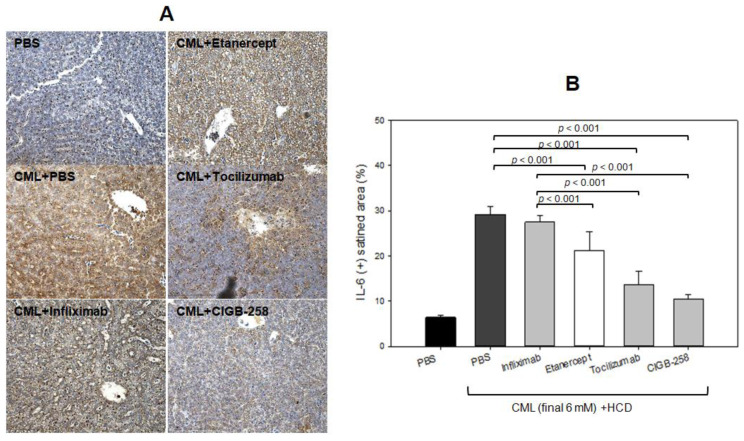
Comparisons of the IL-6-stained area from immunohistochemistry (IHC) among each drug-injected zebrafish. (**A**) Representative image of IHC using IL-6 antibody-stained hepatic tissue. The scale bar indicates 100 μm. (**B**) Quantification of IL-6 antibody-stained area with brown intensity using Image J software version 1.53r (http://rsb.info.nih.gov/ij/ accessed on 14 October 2022).

**Figure 13 ijms-24-07044-f013:**
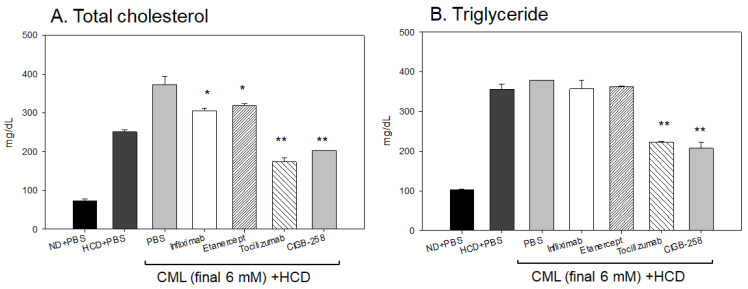
Quantification of lipid profiles from zebrafish injected with PBS, Infliximab, Etanercept, Tocilizumab, and CIGB-258, in the presence of CML under a high cholesterol diet (HCD). *, *p* < 0.05 versus PBS (CML-alone) group; **, *p* < 0.01 versus PBS (CML-alone) group. All zebrafish were sacrificed at 180 min post-injection of each designated drug. (**A**) Quantification of the total cholesterol (TC) levels of zebrafish plasma at 180 min post-injection. (**B**) Quantification of the triglyceride (TG) levels of zebrafish plasma at 180 min post-injection.

## Data Availability

The data used to support the findings of this study are available from the corresponding author upon reasonable request.

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
