# Peer review of "CIGB-258 Exerts Potent Anti-Inflammatory Activity against Carboxymethyllysine-Induced Acute Inflammation in Hyperlipidemic Zebrafish via the Protection of Apolipoprotein A-I"

_ijms, 2023, doi:10.3390/ijms24087044_

Round 1
Reviewer 1 Report
In the manuscript entitled "CIGB-258 exerted potent anti-inflammatory activity in hyper lipidemic zebrafish with carboxymethyllysine-induced acute inflammation via enhancement of apolipoprotein A-I stability" written by Cho et al., the authors showed that CIGB-258, a peptide derived from heat shock protein HSP60, inhibits the extent of apolipoproteinA-I glycation induced with carboxymethyllysine and that a co-microinjection of carboxymethyllysine and CIGB-258 into zebrafish embryos recovers the survivability. Additionally, the authors also showed that an intraperitoneal co-injection of carboxymethyllysine and CIGB-258 produces a faster recovery of the swimming ability and prevents acute death in hyperlipidemic zebrafish. Although the data seem interesting, this reviewer has several concerns.
Major concerns
The molecular weight of apolipoprotein A-I is different between Figures 1B and 2B. In addition, the bands of apolipoprotein A-I in Figure 2B are much fainter than in Figure1B although the applied protein amounts were the same according to Materials and Methods section.
Show error bars in Figure 2A. Is there no statistical significance?
The authors mentioned that multiple groups were compared using a Scheffe test for the zebrafish study. What kind of test was used for apolipoproteinA-I study?
Author Response
Thank you very much heartily for reviewing and critical comments to improve this paper
Please find attached doc file for point-by-point response

Reviewer 2 Report
Recently I agreed to review the manuscript entitled CIGB-258 exerted potent anti-inflammatory activity in hyperlipidemic zebrafish with carboxymethyllysine-induced acute inflammation via enhancement of apolipoprotein A-I stability submitted by
Kyung-Hyun Cho * , Hyo-Seon Nam , Ji-Eun Kim , Hye-Jee Na , Maria Dominguez , Gillian Martinez Donato
During the evaluation of this manuscript, I recognized that the authors recently published a paper in the International Journal of Molecular Sciences entitled Anti-Inflammatory Activity of CIGB-258 against Acute Toxicity of Carboxymethyllysine in Paralyzed Zebrafish via Enhancement of High-Density Lipoproteins Stability and Functionality; https://doi.org/10.3390/ijms231710130.
By checking the relevance of this publication, for me, it became obvious that the authors did not handle this fact in a way they should. Although this former paper (Ref. 21) is listed in the reference list and also in the discussion part (line 484), it is not considered in the rest of the document although it´s content is relevant for the submitted manuscript. Furthermore, the authors did not emphasize how their published results are related to the present study.
From my point of view the normal review procedure is not convenient due to the required extensive revision.
Author Response
Thank you very much heartily for reviewing and critical comments to improve this paper.
Please find attached doc file for point-by-point response.

Reviewer 3 Report
This study examined the protective effect of CIGB-258 for HDL/apoA-I against glycation stress and its anti-inflammatory activity in a hyperlipidemic and hyperinflammatory zebrafish model. This topic is interesting, and this manuscript has a wide approach to answer this goal. Although I recognize the potential of this study, there are some major issues in the design of the experiments and some data are not convincing.
Overall, the methods are poor and do not cover all the methods used in the study. The information on the number of specimens used is missing as well as the description of some analyses such as survivability. The discussion section is good, however, there is some overreach in the conclusions, namely the enhancement of apolipoprotein A-I stability. How did the authors assess the stability? This is not clear. The manuscript requires in depth proofreading, especially in the abstract and introduction.
See below my major concerns:
1) The title should be written in the present tense and should not include a period in the end.
2) This manuscript requires in depth proofreading and language assessment.
3) The abstract is very long and too detailed with some confusing information. Abstracts should be concise and focused on the more important details of the work.
4) In Figure 1, the densiometric analysis of the SDS-Page is represented with the red numbers after the bands? This is merely representative, as it is only one gel. What is the “n” of your experiment? How many biological replicates did you use? In your methods section you state three independent SDS-PAGE. This information should be accounted for in the figure legend and besides the representative gel, the authors should insert a graph with the data including all the independent experiments.
5) In respect to Figure 1B, you state in your methods section: “The apoA-I content was compared by SDS-PAGE and densitometric analysis because the glycation resulted in a severe decrease in the protein content in HDL and several beneficial functions of apoA-I”, however, this decrease is not shown in the SDS-Page, this is an obvious control that is missing.
6) Figure 2a doesn’t have enough resolution for the analysis of the graph. Again, Figure 1B should be accompanied by a graph with all the independent experiments, as it is, the picture and the densiometry values are not convincing.
7) In Figure 3B, the evaluation of ferric ion reduction ability (FRA), the authors mention %s, however, the graph does not show significant differences in absorbance. It is confusing looking at the graph showing absorbances and reading the interpretation as %, especially because this is not translatable in the graph. Furthermore, controls are missing in this experiment: PBS+ CML; PBS+CIGB-258, PBS+CML+ CIGB-258, and PBS+apoA-I+ CIGB-258. These controls will ensure that the observed effects are indeed resulting from the action of CIGB-258 on the ApoA-I and CML condition.
8) Figure 4/5a: How was survivability assessed? This is not described in the methods section as it should. How many embryos were injected and used for these analyses? Again, this information should be stated in the methods section and included in the figure legends.
9) Hematoxylin and Eosin (H&E) staining is a structural staining method, usually used to distinguish differences in the overall tissue and cellular structure. The analysis of the H&E stained area % seems strange to me, as it is a very general observation. Has this analysis been used before in this context? Could the authors provide references for this analysis? Also, the methods section referent to the Histopathological analysis is very poor and unspecific. Oil red O staining is missing from the methods section.
10) How was the swimming ability evaluated? How many individuals were analyzed? How did you calculate the mean of swimming over time? The 9C graph is confusing and not clear. Moreover, this analysis is not described in the methods section and should be carefully defined.
11) In Figure 10A, the authors refer to infiltrated neutrophils. There should be included a higher magnification showing this infiltration. The images are too small for this observation to be clear. Again, the authors use the stained area % as a measurement.
12) The authors state: “These results suggest that CIGB-258 protected hyperlipidemic zebrafish from CML toxicity, such as acute inflammation, paralysis, and death, via stabilization of apo I and HDL.”. The use of the word stabilization puzzles me. Are the authors referring to the levels or some other result that is not shown?
Author Response
Thank you very much heartily for reviewing and critical comments to improve this paper.
Please find attached doc file for point-by-point response

Round 2
Reviewer 1 Report
The authors have addressed all of this reviewer's comments.
Author Response
Thank you very much heartily for reviewing and acceptance of this paper.
Reviewer 2 Report
Comments ijms 2234187-v2
The authors kindly summarized the differences between the two publication, which partially disclose the differences and similarities. Interestingly, the authors only marginally revised their manuscript, however discussion of their previous results is still not included to significantly substantiate this manuscript. It is obvious that the two models are not identical but the relevance of the previous study is evident in particular to demonstrate the extension of the scope.
Published online 2022 Sep 4. doi: 10.3390/ijms231710130
PMCID: PMC9456132
PMID: 36077532
Anti-Inflammatory Activity of CIGB-258 against Acute Toxicity of Carboxymethyllysine in Paralyzed Zebrafish via Enhancement of High-Density Lipoproteins Stability and Functionality
Kyung-Hyun Cho,1,2,* Ji-Eun Kim,1 Hyo-Seon Nam,1 Dae-Jin Kang,1 and Hye-Jee Na1
Background: Hyperinflammation is frequently associated with the chronic pain of autoimmune disease and the acute death of coronavirus disease (COVID-19) via a severe cytokine cascade. CIGB-258 (Jusvinza®), an altered peptide ligand with 3 kDa from heat shock protein 60 (HSP60), inhibits the systemic inflammation and cytokine storm, but the precise mechanism is still unknown. Objective: The protective effect of CIGB-258 against inflammatory stress of N-ε-carboxymethyllysine (CML) was tested to provide mechanistic insight. Methods: CIGB-258 was treated to high-density lipoproteins (HDL) and injected into zebrafish and its embryo to test a putative anti-inflammatory activity under presence of CML. Results: Treatment of CML (final 200 μM) caused remarkable glycation of HDL with severe aggregation of HDL particles to produce dysfunctional HDL, which is associated with a decrease in apolipoprotein A-I stability and lowered paraoxonase activity. Degradation of HDL3 by ferrous ions was attenuated by a co-treatment with CIGB-258 with a red-shift of the Trp fluorescence in HDL. A microinjection of CML (500 ng) into zebrafish embryos resulted in the highest embryo death rate, only 18% of survivability with developmental defects. However, co-injection of CIGB-258 (final 1 ng) caused the remarkable elevation of survivability around 58%, as well as normal developmental speed. An intraperitoneal injection of CML (final 250 μg) into adult zebrafish resulted acute paralysis, sudden death, and laying down on the bottom of the cage with no swimming ability via neurotoxicity and inflammation. However, a co-injection of CIGB-258 (1 μg) resulted in faster recovery of the swimming ability and higher survivability than CML alone injection. The CML alone group showed 49% survivability, while the CIGB-258 group showed 97% survivability (p < 0.001) with a remarkable decrease in hepatic inflammation up to 50%. A comparison of efficacy with CIGB-258, Infliximab (Remsima®), and Tocilizumab (Actemra®) showed that the CIGB-258 group exhibited faster recovery and swimming ability with higher survivability than those of the Infliximab group. The CIGB-258 group and Tocilizumab group showed the highest survivability, the lowest plasma total cholesterol and triglyceride level, and the infiltration of inflammatory cells, such as neutrophils in hepatic tissue. Conclusion: CIGB-258 ameliorated the acute neurotoxicity, paralysis, hyperinflammation, and death induced by CML, resulting in higher survivability in zebrafish and its embryos by enhancing the HDL structure and functionality.
……………………..Major relevant facts are highlighted in red
Author Response
Thank you very much heartily for reviewing and critical comments to improve this paper.
Please find attached doc file for revision letter.
